# Comparative Analysis of Six Complete Plastomes of *Tripterospermum* spp.

**DOI:** 10.3390/ijms25052534

**Published:** 2024-02-22

**Authors:** Xiong-De Tu, Wen-Jun Lin, Hou-Hua Fu, Yi-Zhe Lin, Jun Shen, Shuai Chen, Zhong-Jian Liu, Ming-He Li, Shi-Pin Chen

**Affiliations:** 1College of Forestry, Fujian Agriculture and Forestry University, Fuzhou 350002, China; 2210455001@fafu.edu.cn (X.-D.T.); linwenjun@fafu.edu.cn (W.-J.L.); fhh202216@163.com (H.-H.F.); linyizhe_0220@163.com (Y.-Z.L.); 17859565356@163.com (J.S.); csssd19@163.com (S.C.); zjliu@fafu.edu.cn (Z.-J.L.); 2Key Laboratory of National Forestry and Grassland Administration for Orchid Conservation and Utilization at Landscape Architecture and Arts, Fujian Agriculture and Forestry University, Fuzhou 350002, China

**Keywords:** comparative analysis, Gentianinae, plastid genome, phylogenetic analysis, *Tripterospermum*

## Abstract

The *Tripterospermum*, comprising 34 species, is a genus of Gentianaceae. Members of *Tripterospermum* are mostly perennial, entwined herbs with high medicinal value and rich in iridoids, xanthones, flavonoids, and triterpenes. However, our inadequate understanding of the differences in the plastid genome sequences of *Tripterospermum* species has severely hindered the study of their evolution and phylogeny. Therefore, we first analyzed the 86 Gentianae plastid genomes to explore the phylogenetic relationships within the Gentianae subfamily where *Tripterospermum* is located. Then, we analyzed six plastid genomes of *Tripterospermum*, including two newly sequenced plastid genomes and four previously published plastid genomes, to explore the plastid genomes’ evolution and phylogenetic relationships in the genus *Tripterospermum*. The *Tripterospermum* plastomes have a quadripartite structure and are between 150,929 and 151,350 bp in size. The plastomes of *Tripterospermum* encoding 134 genes were detected, including 86 protein-coding genes (CDS), 37 transfer RNA (tRNA) genes, eight ribosomal RNA (rRNA) genes, and three pseudogenes (*infA*, *rps19*, and *ycf1*). The result of the comparison shows that the *Tripterospermum* plastomes are very conserved, with the total plastome GC content ranging from 37.70% to 37.79%. In repeat sequence analysis, the number of single nucleotide repeats (A/T) varies among the six *Tripterospermum* species, and the identified main long repeat types are forward and palindromic repeats. The degree of conservation is higher at the SC/IR boundary. The regions with the highest divergence in the CDS and the intergenic region (IGS) are *psaI* and *rrn4.5-rrn5*, respectively. The average pi of the CDS and the IGS are only 0.071% and 0.232%, respectively, indicating that the *Tripterospermum* plastomes are highly conserved. Phylogenetic analysis indicated that Gentianinae is divided into two clades, with *Tripterospermum* as a sister to *Sinogeniana*. Phylogenetic trees based on CDS and CDS + IGS combined matrices have strong support in *Tripterospermum*. These findings contribute to the elucidation of the plastid genome evolution of *Tripterospermum* and provide a foundation for further exploration and resource utilization within this genus.

## 1. Introduction

*Tripterospermum* Blume, established by Blume in 1826 with the type species *T. trinerve* from Java, is a genus of Gentianinae (Gentianaceae). There are 34 accepted species of *Tripterospermum*, distributed in East Asia, Southeast Asia, and the adjacent Himalayan region, with the highest diversity in southwest China, and there are more than 20 species [1,2]. The species of *Tripterospermum* are characterized by perennial twining herbs, opposite leaves, and axillary and terminal inflorescences [3]. *Tripterospermum* plants are rich in iridoids, xanthones, flavonoids, and triterpenes, revealing diverse biological activities, such as antivirus and antihypertension [4,5,6,7,8,9].

The subtribe Gentianinae, comprising *Crawfurdia* Wallich, *Gentiana* L., *Kuepferia* Adr. Favre, *Metagentiana* T. N. Ho & S. W. Liu, *Tripterospermum* Blume, and *Sinogentiana* Favre & Yuan, encompasses a total of six genera with approximately 425 species. This subtribe stands as one of the most species-rich clades within Gentianaceae [10,11]. Two genera, *Tripterospermum* and *Crawfurdia*, were concurrently described [12,13]. Due to their climbing habits, these genera are frequently confused [14]. Some authors have even considered them a single genus [15,16,17], while others classified them as *Gentiana* [18,19]. The monophyly of *Tripterospermum* and *Crawfurdia* has been questioned by previous studies [20,21]. Favre et al. [10] employed ITS and a*tpB-rbcL* sequences for nineteen species of *Tripterospermum*, nine of *Crawfurdia*, and eleven of *Metagentiana*. The results indicated that only *Metagentiana* is polyphyletic, whereas *Crawfurdia* and *Tripterosperum* are monophyletic. However, the phylogenetic support within the genus *Tripterospermum* is low. In a subsequent study, Favre et al. [11] utilized ITS, *atpB-rbcL*, and *trnL-trnF* to reconstruct the phylogenetic relationships of Gentianinae. The findings suggested that *Gentiana* sect. *Otophora* is monophyletic, which is more closely related to *Metagentiana* than *Gentiana*. *Metagentiana* is monophyletic when excluding *M. striata* and *M. souliei*. To account for the monophyly of *Gentiana* and *Metagentiana*, new genera were established, *Kuepferia* and *Sinogentiana*, respectively. More recent studies, incorporating ITS, *atpB-rbcL*, and *trnL-trnF,* conducted extensive sampling of Gentianinae, including 30 *Tripterospermum* species, and constructed a phylogenetic tree. The results indicated that *Crawfurdia*, *Gentiana*, *Kuepferia*, *Metagentiana*, *Sinogentiana*, and *Tripterospermum* are all monophyletic, although the phylogenetic support among *Tripterospermum* species remains low [22].

Traditional molecular markers face challenges in resolving phylogenetic relationships at low taxonomic levels, such as *Tripterospermum*, due to the scarcity of informative sites. One effective approach to enhance molecular phylogenetic datasets is to increase the sampling of loci [23]. The plastid genome, characterized by its small size, dense gene content, moderate mutation rate, and abundant site information [24], has proven to be an effective classification tool for seed plants [25,26,27]. It has also been proven to be a valuable tool for intergenus and intragenus classification in the Gentianaceae [28,29,30,31]. In this study, we present two new plastomes of *Tripterospermum*, *T. filicaule* and *T. nienkui*. These newly obtained sequences were compared with four previously described *Tripterospermum* plastomes. Additionally, to further elucidate the phylogenetic relationships within *Tripterospermum* and its subtribe Gentianinae, we conducted a phylogenetic analysis using the published plastid genomes and constructed a phylogenetic tree that demonstrates robust support within *Tripterospermum* and the broader subtribe Gentianinae.

## 2. Results

### 2.1. Characteristics of the Plastome

The plastomes of *T. filicaule* and *T. nienkui* follow the conventional quadripartite arrangement, comprising a sizable single-copy (LSC) region, a compact single-copy (SSC) region, and two inverted repeats (IRs) (Figure 1). The plastome lengths of six *Tripterospermum* species ranged from 150,929 bp (*T. luteoviride*) to 151,350 bp (*T. championii*) (Appendix A). The length of LSC regions ranged from 82,177 bp (*T. luteoviride*) to 82,506 bp (*T. championii*), and the length of the SSC regions ranged from 17,439 bp (*T. nienkui*) to 17,640 bp (*T. championii*). For the IR regions, the sizes ranged from 25,581 bp (*T. japonicum*, *T. membranaceum*, and *T. nienkui*) to 25,602 bp (*T. championii*). The overall GC content of the plastomes ranged from 37.70% to 37.79%, showcasing a noticeable contrast in GC content among the regions. Specifically, the GC content in the IR regions is higher, ranging from 43.42% to 43.47%, compared to the lower GC content observed in the LSC region (35.52% to 35.61%) and SSC region (31.39% to 31.46%) (Appendix A).

The plastomes of six *Tripterospermum* species encoded 134 genes (114 unique), including 86 protein-coding genes (79 unique), 37 transfer RNA (tRNA) genes (30 unique), eight ribosomal RNA (rRNA) genes (four unique), and three pseudogenes (*infA*, *rps19*, and *ycf1*) (Table 1, Appendix A). The examined plastomes revealed an absence of gene organization rearrangements (Appendix A).

Visualization of the SC/IR boundary region in the *Tripterospermum* plastomes revealed a highly conserved pattern (Figure 2). The protein-coding genes *rps19*, *ndhF*, and *ycf1* span the LSC/IRb (JLB), SSC/IRb (JSB), and SSC/IRa (JSA) boundaries. Due to the reverse repetition characteristic of the IR region, *rps19* and *ycf1* generate pseudogenic versions of themselves at IRa and IRb, respectively.

### 2.2. Codon Usage Analysis

A total of seventy-nine unique protein-coding genes (CDS) were analyzed among the six *Tripterospermum* plastomes, including an assessment of codon usage frequency and the calculation of relative synonymous codon usage (RSCU) (Figure 3, Appendix A). The total number of codons for these genes varied between 22,457 (*T. japonicum*) and 22,919 (*T. championii*) (Appendix A) among the six plastomes of *Tripterospermum*. We identified a total of 65 synonymous codons (including stop codons). Among these codons, 32 codons had RSCU ≥ 1, and 33 had RSCU < 1 (Appendix A). Leucine (Leu) was consistently the most frequently encoded amino acid in all *Tripterospermum* plastomes, with a percentage ranging from 10.54% to 10.62%, while cysteine (Cys: 1.15–1.16%) was the least abundant (Figure 3).

### 2.3. Repeat Sequence Analysis

Simple sequence repeats (SSRs) were further identified, and a total of 43, 40, 42, 45, 39, and 39 SSRs were identified in the plastomes of *T. championii*, *T. filicaule*, *T. japonicum*, *T. luteoviride*, *T. membranaceum*, and *T. nienkui*, respectively (Figure 4, Appendix A). Five SSR types, mononucleotides, dinucleotides, trinucleotides, tetranucleotides, and pentanucleotides (mono-, di-, tri-, tetra-, and penta-), all appeared in *Trilterospermum* species, but we did not find hexanucleotides in *T. championii*. Mononucleotide repeats are the most abundant, comprising 62.50%, followed by tetranucleotide repeats at 14.52%, whereas pentanucleotides and hexanucleotide repeats are very rare among these plastomes, accounting for only 2.42% and 2.01% of the total, respectively. In all analyzed plastomes, SSRs are predominantly situated in the LSC and IGS regions (Appendix A).

Four types of long repeats (forward, palindromic, complement, and reverse repeats) were detected in the *Tripterospermum* plastomes. The number of long repeats varied from 14 (*T. filicaule* and *T. luteoviride*) to 29 (*T. championii*) (Appendix A). Forward repeats (56.76% of the total long repeats) and palindromic repeats (34.23% of the total long repeats) are the most abundant in the total repetition and exist in all plastomes. Reverse repeats (8.11% of the total long repeats) are identified in the plastomes of four species, whereas complement repeats (1.00% of the total long repeats) are exclusively found in *T. nienkui* plastomes (Appendix A). We categorized all repeats into three groups based on their length (30–39 bp, 40–49 bp, and 50–59 bp). Among these, ninety-nine (89.19%) have lengths of 30–39 bp, while only six (5.41%) have lengths of 40–49 bp and 50–59 bp.

### 2.4. Sequence Divergence Analysis

The divergence of the six *Tripterospermum* plastomes was assessed using the mVISTA online platform, with *T. championii* as the reference. (Figure 5). The results demonstrated that the full-length *Tripterospermum* plastomes were highly conserved. The coding regions, including exons and t/rRNA regions, are more conserved than non-coding sequences. To gain a more comprehensive insight into sequence divergence among *Tripterospermum* plastomes, both coding regions and intergenic regions were isolated for the calculation of nucleotide variability (Pi) (Figure 6, Appendix A). For coding regions, the mean pi is 0.00071, indicating high conservatism. The highest pi value is in the *psaI* region, but it is only 0.01201. There are 34 regions with a pi value of 0 (39.08% of all coding regions) (Figure 6A, Appendix A). For intergenic regions, the mean pi is 0.00232, indicating high conservatism, and the highest pi value is in the *rrn4.5*-*rrn5* region, with a pi value of 0.01522. There are 14 regions with a pi value of 0 (20.59% of all intergenic regions) (Figure 6B, Appendix A).

### 2.5. Phylogenetic Analyses

To elucidate the phylogenetic relationships of subtribe Gentianae and *Tripterospermum*, we used seventy-nine unique CDSs extracted from the eighty-six plastomes, including eighty-one plastomes from subtribe Gentianinae and five plastomes from Swertiinae, to construct the Gentianae phylogenetic tree (Figure 7). The results showed that each genus of Gentianae is monophyletic and is highly supported, divided into two clades. One clade comprises *Gentiana*, which accounts for the majority and is divided into twelve sections with high support (Figure 7). The other clade includes *Kuepferia*, *Crawfurdia*, *Metagentiana*, *Sinogentiana*, and *Tripterospermum*; *Tripterospermum* is sister to *Sinogentiana*, and towards the base are *Metagentiana*, *Crawfurdia*, and *Kuepferia* (Figure 7). Within the genus *Tripterospermum*, the relationship between the three species, *T. membranaceum*, *T. filicaule*, and *T. nienkui,* has moderate support (Figure 7). To further clarify the impact of using different plastome tree-building strategies on the phylogenetic relationship of *Tripterospermum*, we developed phylogenetic trees based on four datasets: complete plastid genomes, CDS, intergenic regions (IGS), and a combined dataset encompassing both CDS and IGS. The results show that the phylogenetic tree constructed by CDS (Figure 8A) and CDS + IGS (Figure 8C) have similar support, and both are better than IGS (Figure 8B) and complete plastid (Figure 8D) constructed phylogenetic tree, supporting higher and better ability to distinguish *Tripterospermum* species. However, in the genus *Kuepferia*, the phylogenetic tree constructed by CDS has nodes with an unstable relationship (0/45/0.65), while the phylogenetic tree constructed by CDS + IGS had high support (100/100/1.00).

## 3. Discussion

### 3.1. Variation of Plastome Sequences

In this study, we expanded plastome sampling in *Tripterospermum*, providing a valuable opportunity to enhance our understanding of plastome evolution in this complex taxon. We assembled and annotated two *Tripterospermum* plastomes, conforming to the typical quadripartite structure shared by most angiosperms comprising one LSC region, one SSC region, and two IR regions. The size of *Tripterospermum* plastomes varied from 150,929 bp (*T. luteoviride*) to 151,350 bp (*T. championii*). This range aligns with the reported sizes of plastomes in Gentianinae, spanning from 117,780 bp (*Gentiana producta*) to 151,350 bp (*T. championii*) [30]. The six *Tripterospermum* plastomes exhibit conservatism in the number and types of genes, collectively comprising 134 genes, including 86 CDS, 37 tRNAs, and eight rRNAs. The plastid genomes of Gentianeae species have all undergone pseudogenization of the *rps16* gene. Since then, the loss of *rps16*, *ndh* complexes, the second intron of *clpP*, and the intron of *rpl2* have occurred in many genera or *Gentiana* sections in Gentianinae [30]. However, these phenomena were not found in *Tripterospermum*, and only pseudogenization of the *rps16* gene existed.

This study conducted analyses of Mauve collinearity and plastome boundaries, revealing a highly conserved structure among *Tripterospermum* plastomes. The expansion and contraction of the IR boundary play a pivotal role in the evolution of species [32]. Concerning the gene arrangement at the boundaries within the six *Tripterospermum* plastomes, we observed a high degree of conservation at LSC/IRa (JLA), JLB, JSA, and JSB. Due to the spanning of JLB and JSA by the two genes, *rps19* and *ycf1*, their duplicated pseudogenes appear in another IR region, a phenomenon documented in numerous previous studies [28,30,33].

### 3.2. Codon Usage and Repeat Sequence Analysis

The bias in codon usage is a significant factor in the evolution of plastomes and influences the expression of gene functions [34]. Organisms that share close genetic relationships often display highly similar codon usage biases [35]. The patterns of codon frequency and RSCU values were notably similar in *Tripterospermum* plastomes. Among all codons, Leu displayed the highest occurrence (10.54–10.62%), while Cys had the lowest frequency (1.15–1.16%). It is worth noting that the total amount of amino acid in the plastid of *T. filicaule*, *T. membranaceum*, and *T. nienkui* was the same, which was 22,857. *T. filicaule* was a sister to *T. membranaceum*, and they were sisters with *T. nienkui*, and the total amount of amino acid in their plastomes was the same, which is related to their close relationship.

The SSRs, also called microsatellites, comprise repeating 1–6 nucleotide motifs (mono-, di-, tri-, tetra-, penta-, and hexa- repeats). SSRs often exhibit high levels of polymorphism and find extensive applications in species authentication, elucidating evolutionary relationships and assessing genetic diversity within plant populations [36,37,38]. In this study, we identified a total of 39 to 45 SSRs in the plastomes of *Tripterospermum*. We found that the number of SSRs in *Tripterospermum* plastome introns was correlated with phylogenetic relationships. The number of SSRs in the plastome introns of *T. filicaule* was the lowest (2) and located at the top of the phylogenetic tree, followed by *T. membranaceum* (3), *T. nienkui* (3), *T. japonicum* (4), *T. championii* (5), and *T. luteoviride* (5). The plastome introns of *T. championii* and *T. luteoviride* had the highest number of SSRs, located at the base of the phylogenetic tree. Among single-nucleotide repeats, only A/T repeat sequences were observed, with no G/C repeat sequences found. This observation may be attributed to the low occurrence of single-nucleotide repeats and the A/T bias in plastomes [39,40], consistent with previous studies [28]. Furthermore, the numbers of single-nucleotide repeats (A/T) varied across the six *Tripterospermum* plastomes, providing potential utility for future population genetic and phylogenetic studies. Repeat sequences > 30 bp, known as long repeat sequences, were also identified in this study. These sequences may play roles in genome recombination, rearrangement, and phylogeny, and contribute to insertions and substitutions in plastomes [41]. The predominant types of repeats identified were forward repeats and palindromic repeats falling within the 30–39 bp range. These findings are particularly noteworthy for the identification and analysis of genetic diversity in *Tripterospermum* plants.

### 3.3. Sequence Divergence Analysis and Phylogenetic Analysis

To assess the variations of *Tripterospermum* plastomes, mVISTA was employed to investigate the structural characteristics of the six plastomes. The results indicate that coding regions were more conserved than non-coding regions. To explore highly mutated hotspots in *Tripterospermum* plastomes, both coding and intergenic regions were extracted for pi calculation. The mean pi of the coding and intergenic regions was only 0.071% and 0.232%, respectively. This result shows surprisingly low differences between all aligned sequences, highlighting the conservation of coding and intergenic regions, indicating that *Tripterospermum* plastomes are highly conserved. In previous studies, the ITS, *atpB-rbcL*, and *trnL-trnF* sequences could not completely resolve the phylogenetic relationship of *Tripterospermum* at the molecular level [10,11,22]. This requires the introduction of more site sequences.

The phylogenetic relationship within the subtribe Gentianinae, particularly between *Gentiana* and the genera *Crawfurdia*, *Tripterospermum*, and *Metagentiana*, has been a subject of controversy in previous studies [10,15,16,17,18,19,20,21]. To address the non-monophyletic issue of *Gentiana* sect. *Otophora* and *Metagentiana*, Favre et al. [10] established new genera, *Kuepferia* and *Sinogentiana*, respectively. A recent study conducted an extensive sampling study on Gentianinae, revealing that *Crawfurdia*, *Gentiana*, *Kuepferia*, *Metagentiana*, *Sinogentiana*, and *Tripterospermum* are all monophyletic. However, the relationships within the *Tripterospermum* genus are not entirely clear, such as the phylogenies between *Tripterospermum* species being low and requiring further improvement [22].

The significance of plastomes in reconstructing phylogenetic relationships and comprehending evolutionary history has been firmly established, with an expanding application in Gentianinae research. However, existing plastid genome systematic studies in this context only involve a limited number of *Tripterospermum* species [29,30,31]. To explore the potential application of plastid genome systematics in *Tripterospermum*, we utilized 79 unique CDSs extracted from the 86 plastomes to construct a Gentianae phylogenetic tree. The results demonstrated the monophyly of each genus within Gentianae, divided into two well-supported clades. One clade is predominantly represented by *Gentiana*, further divided into twelve sections. The other branch shows *Tripterospermum* as a sister to *Sinogentiana*, followed by *Metagentiana*, *Crawfurdia*, and *Kuepferia*, consistent with Fu et al.’s findings [30]. A short branch-length phylogenetic tree with moderate support is observed in *Tripterospermum*.

To further explore the phylogenetic relationships within *Tripterospermum*, we constructed a phylogenetic tree using four matrices, including complete plastid genomes, CDS, IGS, and CDS + IGS. The trees constructed by CDS and CDS + IGS exhibited similar and higher support compared to those constructed by IGS and the complete plastid matrix and were better at distinguishing *Tripterospermum* species. However, in the genus *Kuepferia*, the phylogenetic tree constructed by CDS showed an unstable relationship node (0/45/0.65), while the tree constructed by CDS + IGS exhibited high support (100/100/1.00). Therefore, CDS and CDS + IGS combined matrices serve as a valuable reference for resolving disputes regarding the phylogenetic relationships of *Tripterospermum*, with the CDS + IGS combined matrix yielding more reliable results overall.

## 4. Materials and Methods

### 4.1. Plant Materials and Sequencing

Fresh leaves of *T. filicaule* and *T. nienkui* were collected from Sanming City, China, and the voucher specimens were deposited at the Herbarium of Fujian Agriculture and Forestry University in Fuzhou, China. Total genomic DNA was extracted from leaves using a modified CTAB method [42], the quantity and quality of extracted DNA was assessed by spectrophotometry, and the integrity was evaluated using a 1% (*w*/*v*) agarose gel electrophoresis. The purified DNA samples were sheared into fragments with an average length of 350 bp for library preparation following the manufacturer’s guidelines (Illumina, San Diego, CA, USA). These libraries were sequenced on the Illumina HiSeq-2500 platform at the Beijing Genomics Institute (Shenzhen, China).

### 4.2. Plastome Assembly and Annotation

The raw reads were verified using Trimmomatic v.0.32 [43] with default settings to obtain high-quality clean reads. The paired-end clean reads were assembled into complete plastid genomes using GetOrganelle v1.7.1 [44] and then annotated by PGA software [45] and were manually adjusted by Geneious 11.1.5 [46]. To avoid confusion of annotation, we re-annotated the plasmids obtained from GenBank. The annotation circle maps of plastomes were drawn using the online tool OGDRAW v.1.3.1 (https://chlorobox.mpimp-golm.mpg.de/OGDraw.html, accessed on 2 October 2023) [47].

### 4.3. Plastome Structure Comparisons and Sequence Divergence Analysis

The rearrangements between different *Tripterospermum* plastomes were identified by Mauve v.2.4.0 [48]. Whole-genome alignment was performed and plotted with the mVISTA program (http://genome.lbl.gov/vista/mvista/submit.shtml, accessed on 17 October 2023) [49] in the Shuffle-LAGAN model. The *T. championii* (MN199139) plastome was used as a reference. The Perl script CPJSdraw.pl (https://github.com/xul962464/CPJSdraw, accessed on 2 November 2023) was used to compare the genes in the boundary regions of LSC/IRb/SSC/IRa. The nucleotide diversity (Pi) of protein-coding genes (CDS) and intergenic regions (IGS) of six *Tripterospermum* plastomes was evaluated using DnaSP v.6.12.01 software [50].

### 4.4. Repeat Sequence and Codon Usage Analysis

The simple sequence repeats (SSRs) were determined by MISA (https://webblast.ipk-gatersleben.de/misa/, accessed on 22 November 2023) [51] with 10, 5, 4, 3, 3, and 3 nucleotide repeats set for mononucleotides (mono-), dinucleotides (di-), trinucleotides (tri-), tetranucleotides (tetra-), pentanucleotides (penta-), and hexanucleotides (hexa-), respectively. Four long repeat types in six plastid genomes, F (forward), P (palindrome), R (reverse), and C (complement) repeats, were detected by the REPuter program (https://bibiserv.cebitec.uni-bielefeld.de/reputer, accessed on 6 November 2023) [52] with a minimum repeat size of 30 bp, an edit distance of three, and 90% similarity. Codon usage and relative synonymous codon usage (RSCU) values were calculated using CodonW v1.4.4 software [53]. Repeat sequences of protein-coding regions were eliminated from the codon usage calculations to avoid sampling errors.

### 4.5. Phylogenetic Analyses

In this study, we used the 81 plastomes of the subtribe Gentianinae to construct phylogenetic trees, including two *Crawfurdia*, sixty-six *Gentiana*, three *Kuepferia*, two *Metagentiana*, two *Sinogentiana,* and six *Tripterospermum*. Five Swertiinae (*Comastoma pulmonarium*, *Gentianopsis paludosa*, *Halenia elliptica*, *Lomatogoniopsis alpina*, and *Swertia erythrosticta*) were set as the outgroups (Appendix A). CDSs were extracted from the 86 plastomes using PhyloSuite v.1.2.2 [54]. By removing repetitive sequences, the final 79 CDS regions were used to construct the phylogenetic tree. The sequences were aligned by MAFFT v7.490 [55] with auto parameters and then concatenated using PhyloSuite v1.2.2 [54].

The phylogenetic trees were inferred by maximum likelihood (ML) and Bayesian inference (BI). ML analyses were performed in IQ-TREE v2.0.3 [56], incorporating the SH-aLRT test and ultrafast bootstrap (UFBoot) feature, with the model identified by ModelFinder implemented in IQ-TREE (-alrt 2000 -B 2000 -m MFP). BI analysis was performed with MrBayes v3.2.7 [57] using best-fit models, which were selected with MrModeltest 2.4 [58] using AIC. The following settings were used: ngen = 5,000,000; samplefreq = 1000; burninfrac = 0.25.

## 5. Conclusions

This study assembled and annotated two *Tripterospermum* plastomes, comparing them with those of other *Tripterospermum* species to investigate plastid genome differences within the genus. The plastomes of *Tripterospermum* displayed a typical quadripartite structure, encompassing an LSC region, an SSC region, and a pair of IRs. The plastome lengths of *Tripterospermum* ranged from 150,929 bp (*T. luteoviride*) to 151,350 bp (*T. championii*) and encoded 134 genes, including 86 protein-coding genes, 37 tRNA genes, eight rRNA genes, and three pseudogenes (*infA*, *rps19*, and *ycf1*). Among these, the CDS region of *psaI* and the IGS region of *rrn4.5*-*rrn5* exhibited the highest variability. Our phylogenetic analysis revealed that matrices constructed using CDS and CDS + IGS provided stronger support within the genus *Tripterygium*. However, for Gentianae, the CDS + IGS combined matrix yielded superior results. This study contributes novel evidence supporting the utility of plastid genomes in elucidating the phylogeny of *Tripterospermum*, thereby aiding in resolving classification challenges within this genus.

## Figures and Tables

**Figure 1 ijms-25-02534-f001:**
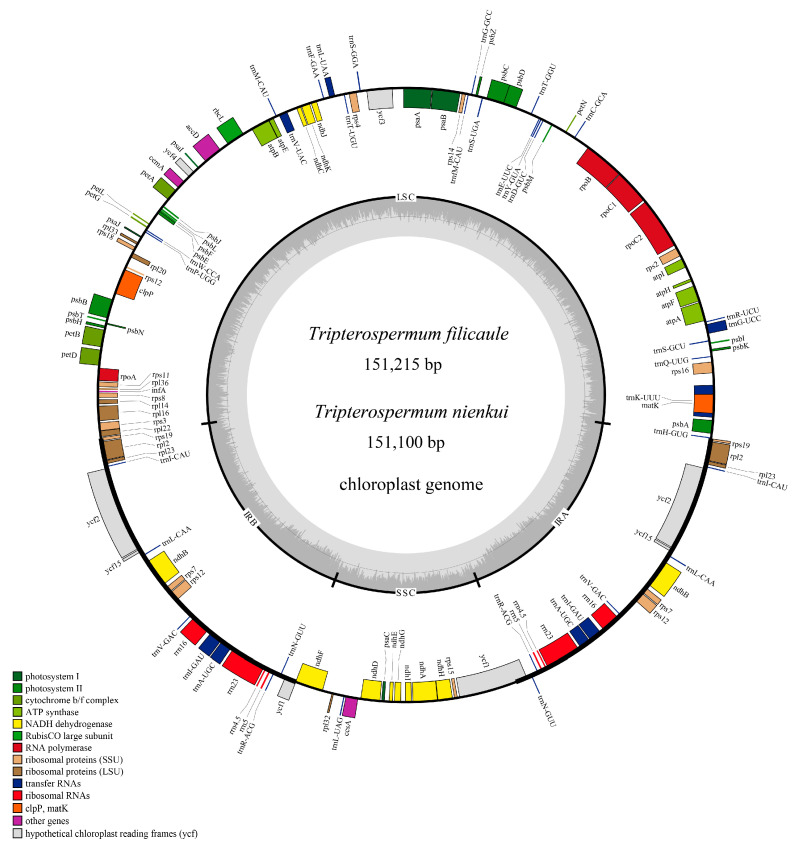
Annotation map of the plastomes for *T. filicaule* and *T. nienkui*. In the inner circle, varying shades of gray depict the distribution of GC content, with darker gray indicating higher GC content. The inner circle further distinguishes between AT content, represented by lighter shades, and GC content, represented by darker shades across the plastome.

**Figure 2 ijms-25-02534-f002:**
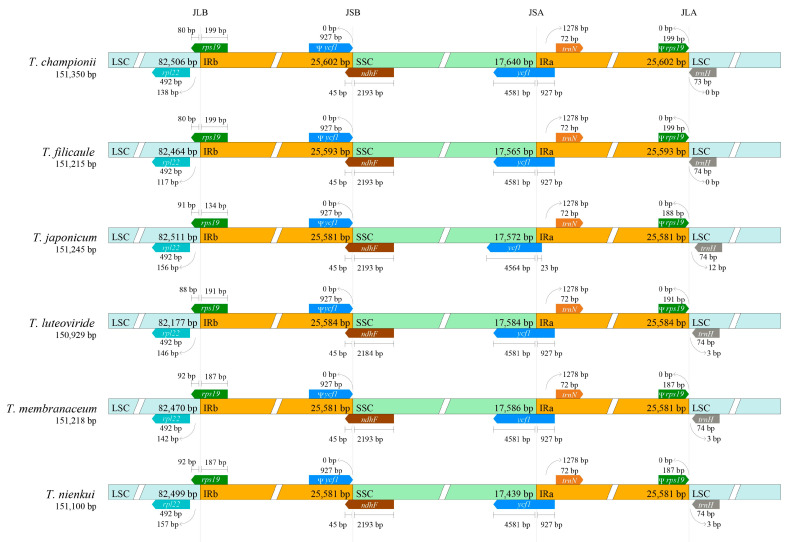
Comparison of junctions among the LSC, SSC, and IRs in the six *Tripterospermum* plastomes. Pseudogenes are marked with (Ψ). The junction sites are labeled as JLB (LSC/IRb), JSB (IRb/SSC), JSA (SSC/IRa), and JLA (IRa/LSC).

**Figure 3 ijms-25-02534-f003:**
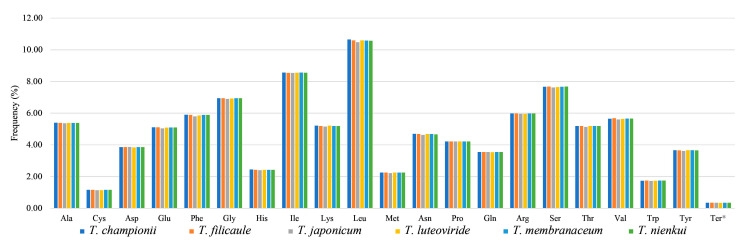
Amino acid frequencies in six *Tripterospermum* plastomes based on protein-coding sequences. * represents the termination codon.

**Figure 4 ijms-25-02534-f004:**
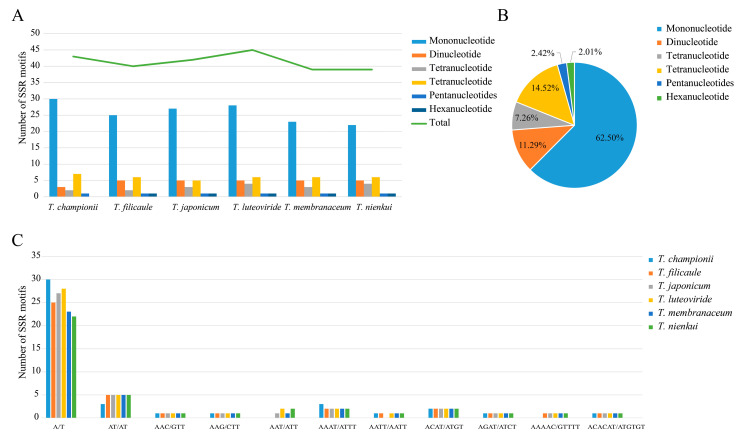
Analyses of SSRs in the six *Tripterospermum* plastomes: (**A**) count and types of SSRs; (**B**) percentage distribution of SSR types; and (**C**) number of SSR motifs.

**Figure 5 ijms-25-02534-f005:**
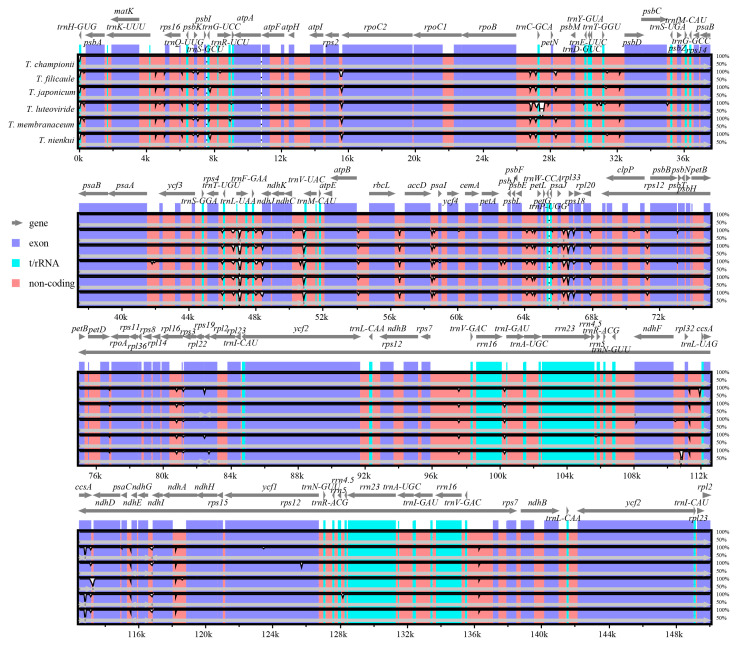
Sequence alignment of six *Tripterospermum* plastomes using the mVISTA program, with *T. championii* serving as the reference. The X- and Y-scales denote the coordinates within plastomes and the percentage of identity (ranging from 50% to 100%), respectively. Gray arrows indicate the transcription direction of each gene. Genome regions are color coded to represent protein-coding regions, tRNA, rRNA, and conserved non-coding regions.

**Figure 6 ijms-25-02534-f006:**
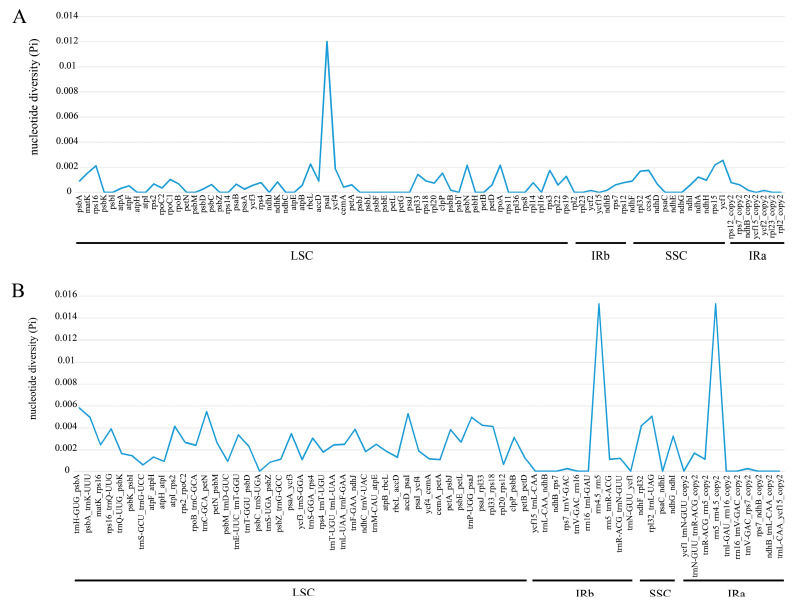
Nucleotide diversity (Pi) of the aligned six *Tripterospermum* plastomes: (**A**) protein-coding genes; (**B**) intergenic regions.

**Figure 7 ijms-25-02534-f007:**
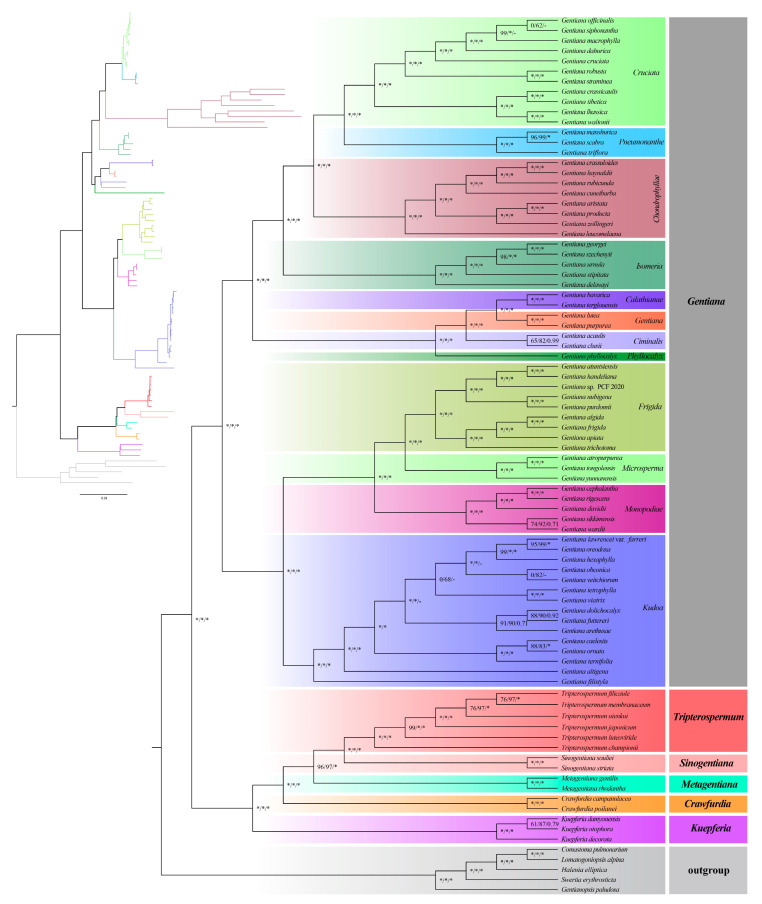
Phylogenetic tree of Gentianinae based on coding genes regions from plastome. Bootstrap percentages and Bayesian posterior probabilities are indicated as values near the nodes (SH-aLRT left, UFBoot middle, and PP right). An asterisk (*) denotes nodes with a confidence level of 100% in bootstrap percentages or 1.00 in Bayesian posterior probabilities.

**Figure 8 ijms-25-02534-f008:**
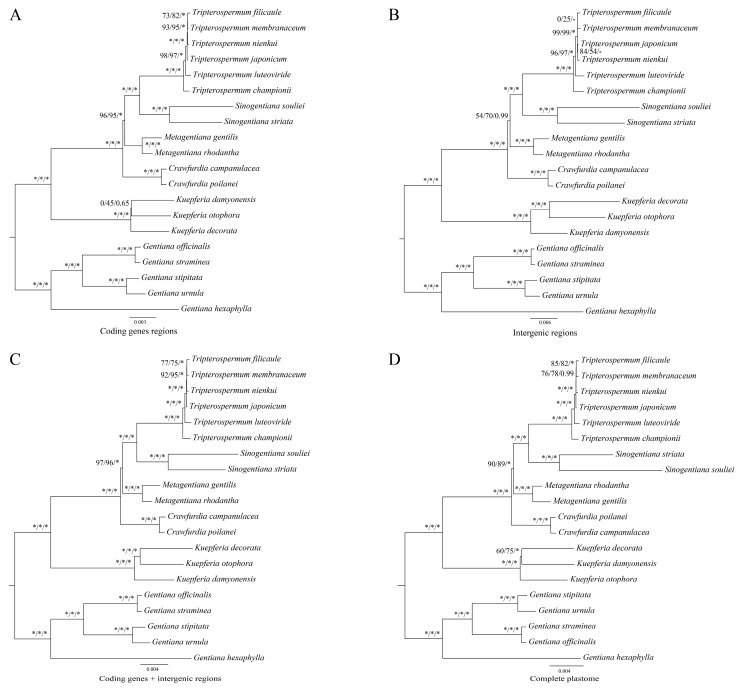
Phylogenetic tree of *Tripterospermum* based on plastome. Bootstrap percentages and Bayesian posterior probabilities are indicated as values near the nodes (SH-aLRT left, UFBoot middle, and PP right). An asterisk (*) denotes nodes with a confidence level of 100% in bootstrap percentages or 1.00 in Bayesian posterior probabilities. (**A**) phylogenetic tree based on coding genes regions (CDS); (**B**) phylogenetic tree based on intergenic regions (IGS); (**C**) phylogenetic tree based on coding genes + intergenic regions (CDS + IGS); (**D**) phylogenetic tree based on complete plastomes.

**Table 1 ijms-25-02534-t001:** Genetic classification of the *Tripterospermum* plastomes.

Category of Genes	Group	Name
Photosynthesis	Photosystem I	*psaA*, *psaB*, *psaC*, *psaI*, *psaJ*
	Photosystem II	*psbA*, *psbB*, *psbC*, *psbD*, *psbE*, *psbF*, *psbH*, *psbI*, *psbJ*, *psbK*, *psbL*, *psbM*, *psbN*, *psbT*, *psbZ*
	Large subunit of rubisco	*rbcL*
	Cytochrome b/f complex	*petA*, *petB **, *petD **, *petG*, *petL*, *petN*
	ATP synthase	*atpA*, *atpB*, *atpE*, *atpF **, *atpH, atpI*
	NADH dehydrogenase	*ndhA **, *ndhB ** (*2), *ndhC*, *ndhD*, *ndhE*, *ndhF*, *ndhG*, *ndhH*, *ndhI*, *ndhJ*, *ndhK*
Self-replication-related genes	Ribosomal RNA	*rrn4*.*5* (*2), *rrn5* (*2), *rrn16* (*2), *rrn23* (*2)
	Transfer RNA	*trnA-UGC ** (*2), *trnC-GCA*, *trnD-GUC*, *trnE-UUC*, *trnF-GAA*, *trnfM-CAU*, *trnG-GCC*, *trnG-UCC **, *trnH-GUG*, *trnI-CAU* (*2), *trnI-GAU ** (*2), *trnK-UUU **, *trnL-CAA* (*2), *trnL-UAA **, *trnL-UAG*, *trnM-CAU*, *trnN-GUU* (*2), *trnP-UGG*, *trnQ-UUG*, *trnR-ACG* (*2), *trnR-UCU*, *trnS-GCU*, *trnS-GGA*, *trnS-UGA*, *trnT-GGU*, *trnT-UGU*, *trnV-GAC* (*2), *trnV-UAC* *, *trnW-CCA*, *trnY-GUA*
	RNA polymerase	*rpoA*, *rpoB*, *rpoC1 **, *rpoC2*
	Small subunit of ribosomal proteins	*rps2*, *rps3*, *rps4*, *rps7* (*2), *rps8*, *rps11*, *rps12 *** (*2), *rps14*, *rps15*, *rps16 **, *rps18*, Ψ*rps19, rps19*
	Large subunit of ribosomal proteins	*rpl2 ** (*2), *rpl14*, *rpl16 **, *rpl20*, *rpl22*, *rpl23* (*2), *rpl32*, *rpl33*, *rpl36*
Other genes	Protease	*clpP* **
	Maturase	*matK*
	Envelop membrane protein	*cemA*
	Acetyl-CoA-carboxylase	*accD*
	Translation initiation factor	Ψ*infA*
	C-type cytochrome synthesis	*ccsA*
Genes with unknown function	Hypothetical chloroplast reading frames	Ψ*ycf*1, *ycf1*, *ycf*2 (*2), *ycf*3 **, *ycf*4

* Gene with one intron, ** gene with two introns, (*2) gene with two copies, and Ψ gene with a pseudogene.

## Data Availability

The two plastome sequences are deposited in GenBank at the NCBI repository with accession numbers OR885001 and OR885002.

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
