# Peer review of "Comparative Analysis of Six Complete Plastomes of Tripterospermum spp."

_ijms, 2024, doi:10.3390/ijms25052534_

Round 1

Reviewer 1 Report

Comments and Suggestions for Authors

The manuscript is nicely presented and well written, organized in a logical way and deals with the comparative plastomes of six  Tripterospermum sp.; it will be more better if authors discuss the evolutionary aspects of the species by correlating the distribution of codon use patten and microsatellites within genome and phylogeny. It will sound interesting.

Best of luck

Comments on the Quality of English Language

Minor improvement in English is necessary.

Author Response

Dear Reviewer,

Thank you very much for your review and valuable suggestions. Following your suggestion, we have revised the manuscript to include a discussion on the evolutionary aspects of Tripterospermum species by correlating the distribution of codon usage patterns, microsatellites within the genome, and phylogeny. This addition aims to provide a more comprehensive understanding of the evolutionary dynamics within the studied species. We have incorporated the recommended changes into the manuscript, and you can find the modified content in the revised sections from line 293 to 380.

We appreciate your insightful comments, which have undoubtedly improved the overall quality of our research. If you have any further feedback or concerns regarding the revised content, please do not hesitate to let us know.

Once again, we are grateful for your time and constructive guidance.

Best regards,

Reviewer 2 Report

Comments and Suggestions for Authors

In this manuscript, Tu et al. present their research on the chloroplastic genomes of the genus Tripterospermum. Specifically, they fully sequenced two new genomes and compared them to four pre-existing ones to gain insight into the structural properties and the phylogenetic affiliations of this genus.

The paper is well written and presented, containing useful and relevant introductory information, robust analysis, and clear presentation. The discussion is at the same level, as well. 

Comments on the Quality of English Language

Very few comments for English:

L. 17-19: Please, rephrase the sentence. "The plastomes of Tripterospermum encoded 134 genes were detected". Maybe by replacing "encoded" with "encoding".

L. 26, "These findings contribute to the plastid genome evolution of Tripterospermum": Pls, rephrase by adding something like: "These findings contribute to the elucidation/examination/study of plastid genome evolution of Tripterospermum".

Author Response

Dear Reviewer,

Thank you for your detailed review and insightful suggestions. We have carefully addressed the minor issues you pointed out:

1. In response to your suggestion for L. 17-19, we have revised the sentence to read: "The plastomes of Tripterospermum, encoding 134 genes, were detected."

2. Regarding L. 26, we have modified the sentence as follows: "These findings contribute to the elucidation of plastid genome evolution in Tripterospermum."

Additionally, we have thoroughly reviewed and proofread the entire manuscript to ensure its overall linguistic accuracy.

We appreciate your time and constructive feedback, which have undoubtedly improved the quality of our work. If you have any further comments or concerns, please do not hesitate to let us know.

Thank you once again for your valuable assistance.

Best regards,

Reviewer 3 Report

Comments and Suggestions for Authors

In this manuscript Xiong‐De Tu and coauthors presented comparative analysis of six complete plastid genomes of plant belonging to the genus Tripterospermum. Plastomes of two species, T. filicaule and T. nienkui, were sequenced for the first time, plastomes of four other species were reported previously (Tripterospermum membranaceum - BMC Plant Biol. 20 (1), 340 (2020); Tripterospermum luteoviride and Tripterospermum championii - Ecol Evol 11 (7), 3286-3299 (2021)). The result of comparison shows that the Tripterospermum plastomes are very similar in size (150,929 and 151,350 bp), structure and are identical gene content. Although this work is technically sound, its novelty and contribution to the field of plant genomics and phylogenetic is very limited and insufficient for publication in a top journal like IJMS. This paper mostly confirmed previously published results and is more appropriate for a specialized like Mitochondrial DNA Resources and something in plant phylogenetics.  

Author Response

Dear Reviewer,

Thank you for your constructive feedback on our manuscript. We appreciate your time and thoughtful evaluation of our work.

We understand your perspective and appreciate your insights. Although there are many shortcomings in the article, we have carefully revised and improved it.

Best regards,

Reviewer 4 Report

Comments and Suggestions for Authors

The current manuscript is interesting but there is some minor revision that should be revised before acceptance.

·       This research entitled “Characteristics and Comparative Analysis of Six Complete Plastomes of Tripterospermum (Gentianinae, Gentianaceae)” has good academic significance. The topic addressed has good scientific depth.

·       Write the important result in the Abstract and revise it, e.g. repeat sequencing analysis, sequencing divergence, and phylogenetic relationships show the main clades.

·       The first paragraph of the introduction starting from (Tripterospermum Blume…… antihypertension), write scientifically and explain the significance of species.

·       Write the objective clear and explanatory.

·       Result section starting first line (The plastomes of T. filicaule and T. nienkui ………..inverted repeats (IRs). These sentences are written below (The plastome lengths of six Tripterosper…… d from 150,929 bp (T. luteoviride) to 151,350 bp).

·       Please, I have attached two important articles related to your studies, follow this article and rephrase the result and abstract section.

·       Sequence divergence Analysis Figure b shows the hotspot regions' genes, written in the result section and abstract.

·       The discussion part should be linked in the result section

·       Material and methods used in the research were not clear and explanatory. Write and explain in each section.

·       Conclusion parts write the significance of the studies. Do not write the result in this part.

1.     Umar Zeb, Azizullah Azizullah, Xiukang Wang, Sajid Fiaz, Hanif Khan. (2021). Comparative genome sequence and phylogenetic analysis of chloroplast for evolutionary relationship among Pinus species. Saudi Journal of Biological Sciences. https://doi.org/10.1371/ journal.

2.     Umar Zeb, Azizullah Azizullah, Sajid fiaz. (2021). Novel insights into Pinus species plastids genome through phylogenetic relationships and repeat sequence analysis. PLOS ONE. https://doi.org/10.1371/ journal.pone.0262040

3.     doi: 10.1111/jse.12492

Please cite and follow these papers. Added more references

Comments on the Quality of English Language

Improve English language

Author Response

Dear Reviewer,

I want to acknowledge the constructive feedback you provided, which we have carefully considered and implemented in the revised version of the manuscript. Your guidance has significantly improved the overall coherence and strength of our study. Your commitment to maintaining the high standards of scholarly review is commendable, and we are truly grateful for your time and expertise. Please find attached the revised manuscript, and we hope that you find the changes reflective of your constructive feedback.

If you have any further suggestions or concerns, please do not hesitate to reach out. We highly value your expertise and insights.

Once again, thank you for your invaluable contributions to the improvement of our manuscript.

Best regards,

Reviewer 5 Report

Comments and Suggestions for Authors

Comments and Suggestions for Authors

In the present manuscript, two new plastomes of the species T. filicaule and T. nienkui from the genus Tripterospermum, Gentianaceae are presented and the results of their comparison with four previously described plastomes of Tripterospermum were еxposed in order to its delimitation from the genus Crawfurdia.

The genera Tripterospermum and Crawfurdia of the Gentianaceae have been described at the same time but are often confused due to their climbing habits. They have even been considered a single genus or classified as Gentiana. The plastid genome, characterized by its small size, dense gene content, moderate mutation rate, and abundant site information, has proven to be a valuable tool for intergeneric and intrageneric delineation. In the study, a phylogenetic analysis was therefore performed using the published plastid genomes and a phylogenetic tree was constructed.

The findings of the study contribute to the plastid genome evolution of Tripterospermum, and provide a foundation for further exploration and resource utilization within this genus.

The manuscript is arainged according to Author guidlines of “International Journal of Molecular Sciences”. The research methods applied are appropriate and sufficient to achieve the objectives of the study. The presenting of results is well structured and supported by figures that are of good quality, and by statistical analysis.

The following recommendations can be made:

In the captures of figures (including the supplementary) to use the short rather than the full article: Figure 1. The annotation map …”- Figure 1. Annotation map …, “Figure 2. The comparison of ..” – Figure 2. Comparison of .., and etc.

In the footnote of Table 1 the signs in parentheses to be placed before the explanation: “Genes marked  with one intron, (**)genes marked  with two introns, (*2) genes marked with two copies, (Ψ) pseudogenes.

In conclusion, this manuscript is recommended for publication in “International Journal of Molecular Sciences”.

Author Response

Dear Reviewer,

I am writing to express my sincere appreciation for your thorough review of our manuscript, "Characteristics and Comparative Analysis of Six Complete Plastomes of Tripterospermum (Gentianinae, Gentianaceae)" and for the valuable feedback you provided.

Your insightful comments and suggestions have been immensely beneficial in refining the content and structure of our paper. I want to inform you that I have carefully considered each of your recommendations and have made the necessary revisions accordingly. The attached document includes a detailed overview of the modifications made throughout the manuscript.

If you have any additional comments or if there are further aspects you would like me to address, please feel free to let me know. Your expertise is highly valued, and your feedback has been instrumental in shaping the final version of our manuscript.

Once again, thank you for your time, commitment, and valuable insights.

Best regards,

Round 2

Reviewer 3 Report

Comments and Suggestions for Authors

In the revised version of this manuscript the authors made some modifications, mostly in the abstract. However, such modificationы cannot change the main content of this study. I can only repeat my conclusion and recommendation: although this work is technically sound, its novelty and contribution to the field of plant genomics and phylogenetic is very limited and insufficient for publication in a top journal like IJMS. This paper mostly confirmed previously published results and is more appropriate for a specialized journal like Mitochondrial DNA Resources or in a journal focused on plant phylogenetics.

Author Response

Thank you very much for your feedback. I have made every effort to revise it